# Experimental Modeling of Natural Processes of Nepheline Alteration

**Julia A. Mikhailova** [1,2,*], **Galina O. Kalashnikova** [2], **Yakov A. Pakhomovsky** [1,2], **Ekaterina A. Selivanova** [1,2] and **Alena A. Kompanchenko** [1]

[1] Geological Institute, Kola Science Centre, Russian Academy of Sciences, 184209 Apatity, Russia; pakhom@geoksc.apatity.ru (Y.A.P.); selivanova@geoksc.apatity.ru (E.A.S.); a.kompanchenko@ksc.ru (A.A.K.)

[2] Nanomaterials Research Centre, Kola Science Centre, Russian Academy of Sciences, 184209 Apatity, Russia; g.kalashnikova@ksc.ru

[*] Correspondence: j.mikhailova@ksc.ru; Tel.: +7-81555-79333

**Abstract:** Nepheline, ideally $Na_3K(Al_4Si_4O_{16})$ is a key mineral of silica-undersaturated igneous rocks. Under subsolidus conditions, nepheline is intensively replaced by numerous secondary minerals, of which various zeolites (mainly natrolite, analcime, gonnardite), as well as cancrinite, muscovite and Al-O-H phases (gibbsite, böhmite, nordstrandite) are the most common. In the rocks of the Lovozero alkaline massif (Kola Peninsula, NW Russia), nepheline is extensively replaced by the association natrolite + nordstrandite ± böhmite ± paranatrolite. To reproduce the conditions for the formation of such a mineral association, a series of experiments were carried out on the dissolution of nepheline in deionized water, 0.5 mol/L NaCl, 0.5 mol/L NaOH, and 0.1 mol/L HCl at 230 °C for 1/5/15 days. When nepheline is partially dissolved, phases and mixtures of phases precipitate on the surface of its grains, and these phases were diagnosed using X-ray powder diffraction and Raman spectroscopy. Observations in natural samples and experimental studies have shown that the nepheline alteration in the rocks of the Lovozero massif with the formation of natrolite and Al-O-H phases occurred under the influence of a high to medium salinity solution at a pH of near 6.

**Keywords:** nepheline; Lovozero massif; hydrothermal experiments; alteration

## 1. Introduction

Nepheline, ideally $Na_3K(Al_4Si_4O_{16})$, is a silica-poor aluminosilicate of the feldspathoid group. The nepheline crystal structure is a derivative of tridymite-type framework, where the voids are filled with Na and K atoms, and half of the Si atoms are replaced by aluminum [1–3]. Nepheline is a key mineral of silica-undersaturated (quartz-free) igneous rocks and related pegmatoid rocks, e.g., Lovozero and Khibiny massifs at Kola Peninsula, Russia; Ilímaussaq massif in Greenland [4–7].

Under subsolidus conditions, nepheline is intensively replaced by various secondary minerals, of which various zeolites (mainly natrolite, analcime, gonnardite), as well as cancrinite, muscovite and Al-O-H phases (gibbsite, böhmite, nordstrandite) are the most common [8–13]. Both the associations of secondary minerals and the intensity of substitutions depend on the composition of the hydrothermal solution affecting the nepheline. For example, in Fohberg phonolite (Kaiserstuhl Volcanic Complex, Germany), zeolites replace nepheline during subsolidus hydrothermal alteration (<150 °C) under alkaline conditions. A sequence of Ca–Na-dominated zeolite species (gonnardite, thomsonite, mesolite) is followed by natrolite. Such sequence reflects changes in aqueous cation ($Ca^{2+}$, $Na^+$) to hydrogen ion activity ratios [9]. In nepheline syenite from the Bang Phuc massif (NE Vietnam), nepheline is replaced by cancrinite and zeolites (natrolite and subordinate analcime), accompanied by minor amounts of dawsonite, nordstrandite, and muscovite. Cancrinite was formed as a result of the reaction nepheline + calcite + $nH_2O$. The presence

of dawsonite indicates high $CO_2$ partial pressure and the presence of high to neutral pH alkaline fluids [14].

In our previous work [12], we studied the products of nepheline alteration in the rocks of the Lovozero massif. It was found that this mineral is extensively replaced by the natrolite + nordstrandite ± böhmite ± paranatrolite association and proposed the following substitution schemes: 3Nph + 4H₂O → Ntr + Nsd + NaOH; 6Nph + 9H₂O → Ntr + Pntr + 2Nsd + 2NaOH, where Nph is nepheline, Ntr is natrolite, Nsd is nordstrandite, and Pntr is paranatrolite. The present article is a continuation of the studies of secondary alterations of nepheline. Here we present the results of a series of experiments aimed at imitating the natural processes of hydrothermal alteration of nepheline. Unaltered nepheline from the Lovozero massif was chosen as the material for the study. We were able to reproduce the mineral associations observed in natural samples and evaluate the change in the composition of the hydrothermal solution.

## 2. Short Geological Backgrounds and Natural Prototype for Experiments

The Lovozero massif with an area of 650 km² is located on the Kola Peninsula, Russia (Figure 1a). This is the second largest by area after the Khibiny massif (ca. 1300 km²) in the world. Lovozero massif is a layered laccolith-type intrusion was emplaced 360–370 Ma ago [15,16] into Archean granite gneisses covered by Devonian volcaniclastic rocks [17,18]. It consists of three major units or complexes [4,19,20] (Figure 1b):

(1) The Layered complex (77% of the massif volume) consists of numerous sub-horizontal layers (or rhythms). Each rhythm is a sequence of following rocks (from top to bottom): trachytoid meso- to melanocratic nepheline syenite (lujavrite)—trachytoid to massif leucocratic nepheline syenite (foyaite)—foidolite (urtite or ijolite).

(2) The Eudialyte complex (18% of the massifs volume) overlaps the Layered complex. This complex is not layered and consists of lujavrite enriched in eudialyte-group minerals.

(3) The Poikilitic complex (5% of the massifs volume) consists of leucocratic feldspathoid syenites, in which grains of feldspathoids are poikilitically incorporated into large crystals of alkali feldspar.

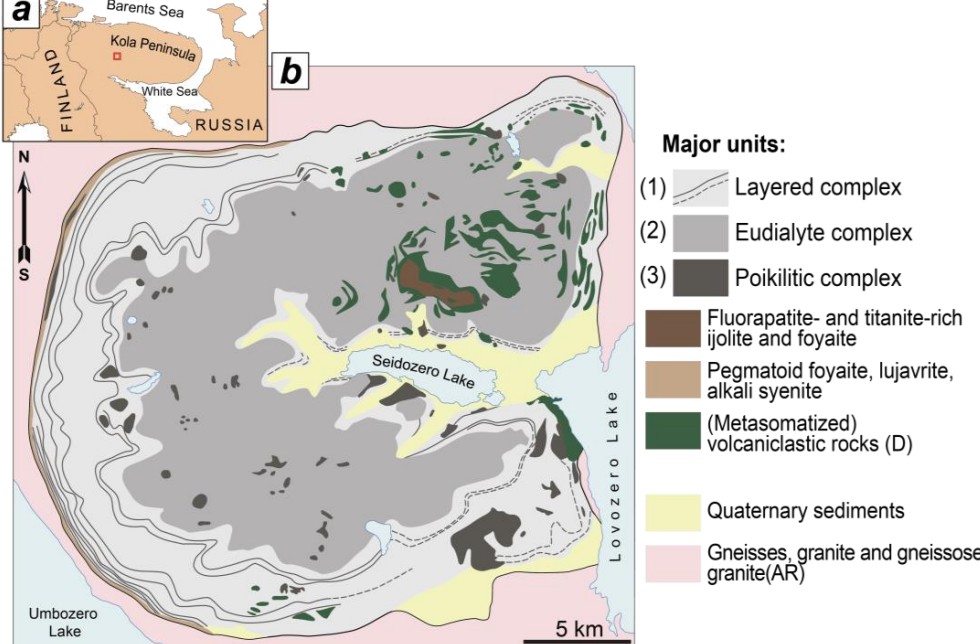

**Figure 1.** Geological background. (**a**) location of the Lovozero massif (red square); (**b**) geological scheme of the Lovozero massif (after [20] with simplifications).

Thus, the bulk of the Lovozero massif is composed of nepheline syenites and foidolites, i.e., nepheline-rich rocks. According to I. V. Bussen and co-authors [21], nepheline composes 20%–25% of the total massif volume, while in some rock varieties, for example, in urtite of the Layered complex, the modal content of nepheline can reach 95 vol.%. Nepheline is an early-magmatic mineral and usually forms euhedral to subhedral grains 1–5 mm across [4,5,21]. Nepheline grains are usually not zonal; the main impurity is iron (both ferrous and ferric). This mineral often hosts small needle-like aegirine inclusions. Wherein, the total iron content in nepheline saturated with aegirine needles is approximately an order of magnitude less than in nepheline without aegirine inclusions [22]. Aegirine needles are usually associated with gaseous inclusions, consisting mainly of methane and hydrogen. Rounded gas bubbles are often attached to aegirine needles or located in secondary trails along healed cracks [23–25].

The ubiquitous intensive substitution of Lovozero' nepheline by zeolites, mainly natrolite, was previously noted by many researchers [4,5,21,26,27]. However, our detailed studies of nepheline substitution products by microtextural, microprobe, and spectroscopic methods showed that Al-O-H phases, namely nordstrandite and/or böhmite, are constantly present in close intergrowths with natrolite (Figure 2a–d). It was also found that the volume ratio of zeolites and Al-O-H phases changes very slightly and is equal to 5.25/1. Based on these observations, the following nepheline substitution reaction was proposed [12]:

$$3NaAlSiO_4 \text{ (Nph)} + 4H_2O \rightarrow Na_2Al_2Si_3O_{10} \cdot 2H_2O \text{ (Ntr)} + Al(OH)_3 \text{ (Nsd)} + NaOH \quad (1)$$

where Nph is nepheline, Ntr is natrolite, Nsd is nordstrandite.

Fine-grained aggregates of secondary minerals are often loose, contain numerous cracks, and are in places macroscopically colored black. According to Raman spectroscopy, the black color is due to the presence of carbon materials (Figure 2e) [28].

Pseudomorphized samples of nepheline, in particular, presented in Figure 1, were natural prototypes for this experimental work.

Studies of natural samples have shown that the alteration of nepheline by an association of zeolites and Al-O-H phases occurs throughout the volume of the Lovozero massif. Indeed, intensive nepheline alterations are observed in samples from the well cores. Additionally, the alteration of nepheline occurs in near-surface conditions. On the surface of outcrops of nepheline-bearing rocks in the Lovozero massif, crusts of thermonatrite constantly appear, which is formed through the following successive reactions [12]:

$$3NaAlSiO_4 \text{ (Nph)} + 4H_2O \rightarrow Na_2Al_2Si_3O_{10} \cdot 2H_2O \text{ (Ntr)} + Al(OH)_3 \text{ (Nsd)} + NaOH \quad (2)$$

$$NaOH + CO_2 \text{ (in air)} \rightarrow Na_2(CO_3) \cdot H_2O \text{ (Tnat)} \quad (3)$$

where Nph is nepheline, Ntr is natrolite, Nsd is nordstrandite, Tnat is thermonatrite.

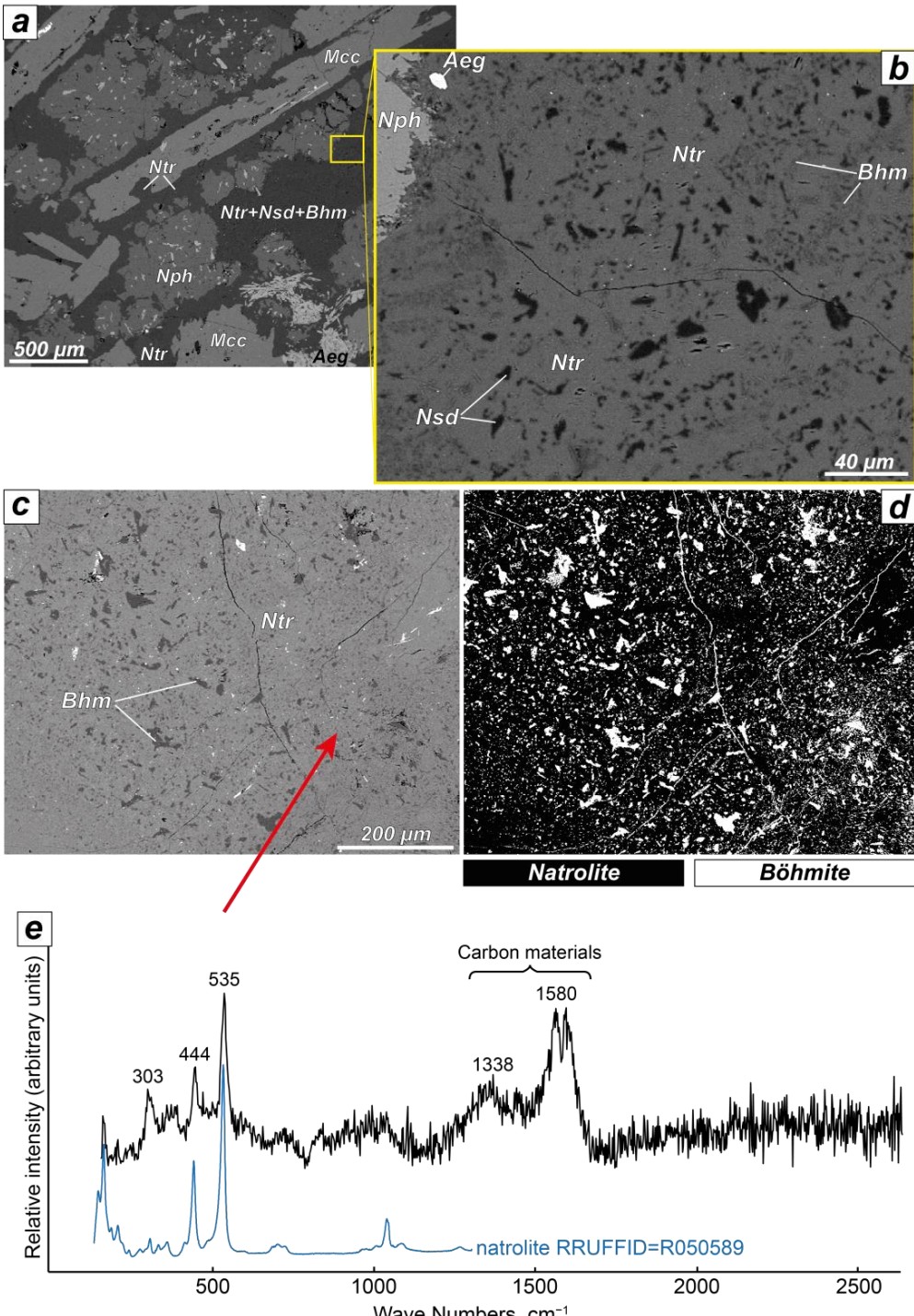

**Figure 2.** Secondary alteration of nepheline (Nph) in rocks of the Lovozero massif. (**a**) replacement of nepheline by an aggregate of natrolite (Ntr), nordstrandite (Nsd), and böhmite (Bhm) in foyaite of the Layered complex (after [12] with modifications); (**b**) detailed BSE image of (**a**); (**c**) nepheline alteration products; (**d**) binary pattern of (**c**), in which black indicates natrolite (82.3% of secondary minerals area), and white indicates böhmite (17.7% of secondary minerals area); (**e**) Raman spectrum of natrolite coated (?) with a film of carbon materials; The black line is the spectrum obtained in this study; the colored line is the spectrum from the RRUFF database (https://rruff.info accessed on 19 July 2023). The red arrow indicates the analysis point. (**a**–**d**) Back-scattered electron (BSE) images. Aeg—aegirine, Mcc—microcline.

### 3. Materials and Methods

*3.1. Materials and Design of the Experimental Study*

For this study, an urtite sample (LV-01-45) from the Layered complex of the Lovozero massif containing unaltered nepheline was selected. From this sample, 7 g of pure nepheline was picked; grain size ranged from 1 to 3 mm across. Then the nepheline grains were cleaned in an ultrasonic bath in distilled water to remove submicron-sized particles that adhere to the nepheline grain surface.

The general scheme of the study is shown in Figure A1.

*3.2. Experimental Conditions*

Four series of hydrothermal experiments were carried out, differing in the composition of the initial solution. The experimental conditions are summarized in Table 1. For the experiments, deionized water, 0.5 mol/L NaCl, 0.5 mol/L NaOH, and 0.1 mol/L HCl of reagent-grade quality (Neva reactive) were used. Nepheline grains (about 0.4000 g) were kept in solution (25 mL) at 230 °C for 1/5/15 days without periodic shaking in the PTFE-lined hermetically sealed autoclaves for hydrothermal synthesis (TOPH, Republic of Korea). The ratio of the volume of the autoclave to the volume of the solution was 1.6. During the experiments, the pressures in the autoclaves were autogenous pressures. An ED224S-RCE Sartorius analytical scale (Göttingen, Germany) was used for weighing. After each experiment, the nepheline grains were washed twice with deionized water (30 mL) and dried in air for 1 h.

**Table 1.** Experimental conditions.

| Series of Experiments | Solution | pH | Experiment | Mass of Nepheline Grains, g | Time, Days |
|---|---|---|---|---|---|
| 1 | deionized $H_2O$ | 6.1 | Aks. 4 | 0.4000 | 1 |
| | | | Aks. 8 | 0.4017 | 5 |
| | | | Aks. 12 | 0.4066 | 15 |
| 2 | 0.5 mol/L NaCl | 5.6 | Aks. 2 | 0.4005 | 1 |
| | | | Aks. 6 | 0.4051 | 5 |
| | | | Aks. 10 | 0.4030 | 15 |
| 3 | 0.1 mol/L HCl | 3.0 | Aks. 1 | 0.4037 | 1 |
| | | | Aks. 5 | 0.4000 | 5 |
| | | | Aks. 9 | 0.4012 | 15 |
| 4 | 0.5 mol/L NaOH | 7.8 | Aks. 3 | 0.4033 | 1 |
| | | | Aks. 7 | 0.4036 | 5 |
| | | | Aks. 11 | 0.4037 | 15 |

*3.3. Methods*

The methods used in this work are summarized in Table 2.

**Table 2.** Methods and equipment.

| | Method | Equipment and Analysis Conditions | Equipment Location |
|---|---|---|---|
| Chemical composition of unaltered nepheline | Electron Microprobe analysis | Cameca MS-46 electron microprobe (Cameca, Gennevilliers, France); WDS-mode at 22 kV; beam diameter 10 μm; beam current 20–40 nA; counting times 10 s (for a peak) and 10 s (for background before and after the peak); 5–10 counts for every element in each grain. Standards: lorenzenite (Na), pyrope (Al), wollastonite (Si, Ca), wadeite (K), hematite (Fe). The analytical precision (reproducibility): 0.2–0.05 wt% (2 standard deviations) for the major element; 0.01 wt% for impurities. The systematic errors were within the random errors. | GI KSC RAS |
| | Wet chemical analysis | The accuracy limits for all components are 0.01 wt%. The analysis procedure is as follows: (1) nepheline was dissolved in weak HCl, (2) the insoluble residue was removed and (3) the composition of the solution was analyzed. | |
| Morphology of nepheline alteration products | Scanning electron microscopy | Scanning electron microscope LEO-1450 (Carl Zeiss Microscopy, Oberkochen, Germany) with the energy-dispersive system AZtec UltimMax 100 (Oxford Instruments, Abingdon, UK) | GI KSC RAS |
| Diagnosis of nepheline substitution products | Raman Spectroscopy | EnSpectr R532 (Spectr-M, ISSP RAS, Chernogolovka, Russia) spectrometer equipped with an Olympus BX-43 microscope. Solid-state laser (532 nm) with an actual power of 18 mW under the 50× objective (NA 0.4). The spectra were obtained in the range of 70–4000 $cm^{-1}$ at a resolution of 5–8 $cm^{-1}$ at room temperature. The number of acquisitions is 20. All spectra were processed using the algorithms implemented in the OriginPro 8.1 software package (Originlab Corporation, Northampton, MA, USA). | Mining Institute KSC RAS |
| | Powder X-ray Diffraction | URS-1 powder diffractometer operated at 40 kV and 16 mA with RKU-114.7 mm camera and FeKα-radiation | GI KSC RAS |
| Compositions of the solutions | Inductively coupled plasma mass spectrometry (ICP-MS) | ELAN 9000 DRC-e (Perkin Elmer, Waltham, MA, USA) | Institute of North Industrial Ecology Problems KSC RAS |
| pH of the solutions | | AMT28F pH meter (Hanna, Germany); the admissible error is +/−0.1 pH. | GI KSC RAS |

## 4. Results

*4.1. Nepheline before the Experiment (Sample LV-01-45): Morphology and Chemical Composition*

Sample LV-01-45 is urtite from the upper part of the Layered complex of the Lovozero massif. The modal content of nepheline in this sample is 90 vol.%. Nepheline forms polygonal grains 0.4–2.2 mm across (Figure 3a). In addition to nepheline, anhedral aegirine and magnesio-arfvedsonite, as well as single fine grains of fluorapatite were also found. As a rule, nepheline grains do not contain any inclusions; however, numerous small aegirine needles are present in the central parts of some grains (Figure 3b). Almost every aegirine needle crystal has a gas bubble attached to it (Figure 3c,d). On the Raman spectra, nepheline showed characteristic peaks at 991–995 cm$^{-1}$, 467–469 cm$^{-1}$, 399–402 cm$^{-1}$ and 209–214 cm$^{-1}$. According to the results of Raman spectroscopy, gas inclusions are composed of methane (Figure 3e) [29].

The nepheline chemical compositions according to microprobe and wet chemistry analysis are shown in Table 3. According to microprobe analysis, the formula for nepheline (based on 16 oxygen atoms) is as follows:

$$Na_{2.97}K_{0.59}Al_{3.64}Fe^{3+}_{0.10}Si_{4.30}O_{16}.$$

**Table 3.** Chemical composition of nepheline from sample LV-01-45.

| Component, wt.% | Method | |
|---|---|---|
| | **Microprobe** | **Wet Chemistry** |
| $SiO_2$ | 44.62 | 44.80 |
| $Al_2O_3$ | 32.05 | 31.94 |
| $Fe_2O_3$ | 1.33 | 0.98 |
| FeO | - | 0.30 |
| $Na_2O$ | 15.90 | 15.84 |
| $K_2O$ | 4.76 | 4.62 |
| $H_2O$ | - | - |
| Loi | - | 1.42 |
| Total | 98.66 | 99.90 |

Loi—loss on ignition.

The nepheline formula, calculated based on the results of wet chemical analysis, is as follows (O = 16): $Na_{2.98}K_{0.59}Al_{3.65}Fe^{3+}_{0.07}Fe^{2+}_{0.02}Si_{4.31}O_{16}$

This coincides with the formula calculated from the data of microprobe analysis.

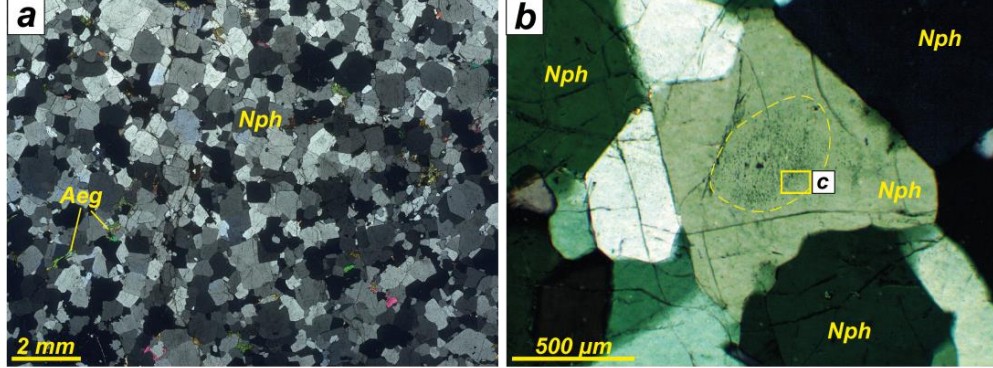

**Figure 3.** *Cont.*

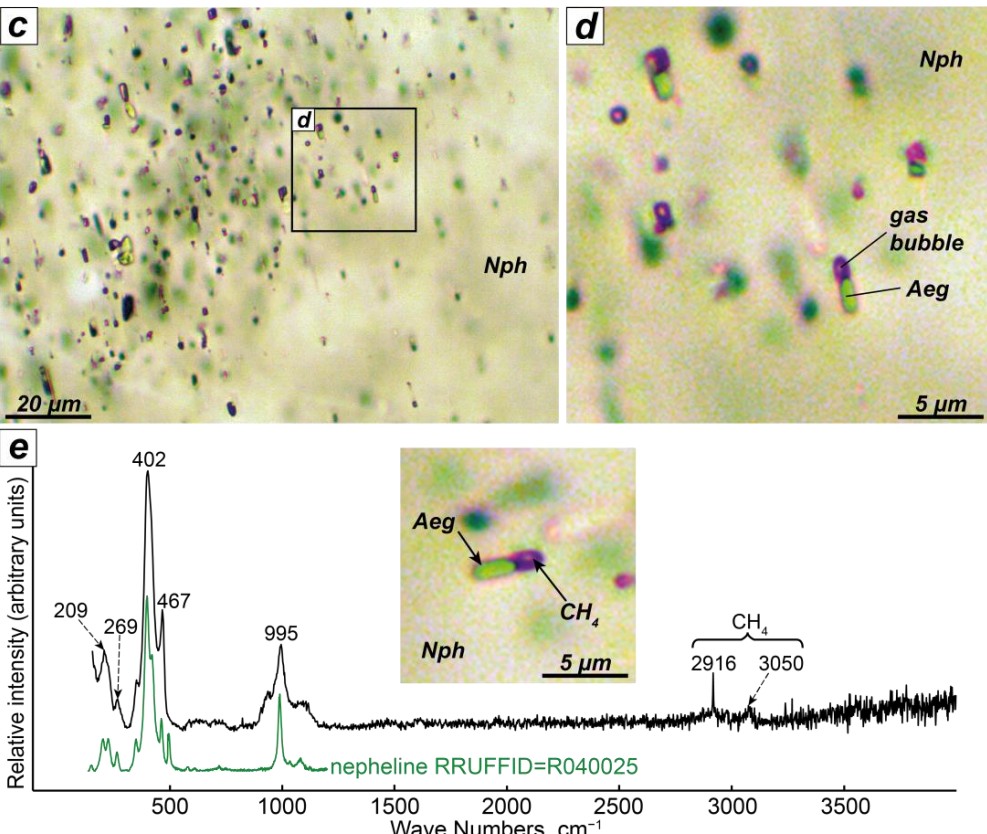

**Figure 3.** Nepheline before the experiment (sample LV-01-45). (**a**) polygonal nepheline grains in urtite LV-01-45; (**b**) nepheline grain with inclusions in the central part (the area enriched with inclusions is outlined by a yellow dotted line); neighboring nepheline grains do not contain inclusions; (**c**) detailed photo of a fragment of a nepheline grain saturated with inclusions; (**d**) inclusions in nepheline, consisting of an aegirine crystals and a gas bubbles attached to it; (**a**–**d**) photos of a polished thin section in polarized light; (**e**) Raman spectrum of a gas inclusion in nepheline. The black line is the spectrum obtained in this study; the colored line is the spectrum from the RRUFF database (https://rruff.info accessed on 19 July 2023). Aeg—aegirine, Nph—nepheline.

### 4.2. Nepheline after the Experiments: Changes in Mass and Color

During all experiments, changes in the mass of nepheline were detected, and during experiments lasting 5 and 15 days, changes in the color of nepheline grains occurred (Figure 4). In experiments with deionized water, the initial mass decreased by 1.9% in 1 day, and in 15 days the mass decreased by 2.8%. In 5- and 15-day experiments, nepheline grains became matte with a brown tint due to the precipitation of the finest crusts of secondary phases.

In experiments with 0.5 mol/L NaCl, the mass of nepheline decreased by 2.4% in 1 day, while the decrease in mass in an experiment lasting 5 days was only 0.8%. This is probably due to the intensive precipitation of secondary phases on the surface of nepheline grains. Indeed, in the 5-day experiment, the nepheline grains became dull with a brown tint, and after the 15-day experiment, the nepheline grains became light brown.

In experiments with 0.1 mol/L HCl, the mass loss after 1 day was 1.0%, while in the experiment lasting 5 days, the mass loss was 5.1% of the initial one. In the experiment with 0.1 mol/l HCl for 15 days, the weight loss significantly decreased, apparently due to the precipitation of secondary phases on the surface of the nepheline grains. After a 15-day experiment, the nepheline grains were covered with the thinnest white crusts.

In experiments with 0.5 mol/L NaOH lasting 1, 5, and 15 days, the mass loss increased sequentially from 0.1 to 4.7% of the initial mass. Small light brown spots were found on the surface of nepheline grains after a 5-day experiment, while after an experiment lasting 15 days, nepheline grains have a distinct brown tint.

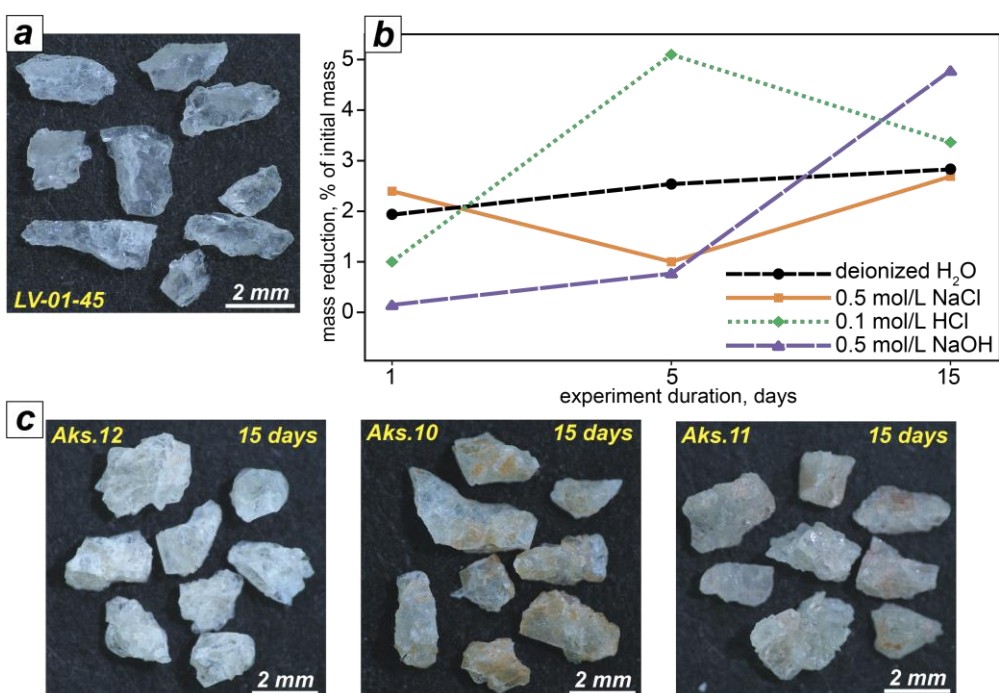

**Figure 4.** Change in mass and color of nepheline grains after experiments. (**a**) nepheline grains before experiments (sample LV-01-45); (**b**) mass reduction after experiments; (**c**) examples of color change of nepheline grains after experiments with deionized water (Aks. 12), 0.5 mol/L NaCl (Aks. 10) and 0.5 mol/L NaOH (Aks. 11). (**a,c**) macroscopic photos (camera Canon PowerShot G9).

### 4.3. Nepheline after Experiments: New Phases on Grain Surfaces

During the experiments, thin crusts of various phases were precipitated on the surface of nepheline grains. Since the newly formed phases are very small, their diagnosis was carried out both by X-ray powder diffraction and by Raman spectroscopy. Figures 5–8 show BSE-images and photos of the precipitated phases and the corresponding Raman spectra, and Tables 4–7 show the results of X-ray studies. Table 8 lists the phases precipitated on the surface of partially dissolved nepheline grains after all experiments.

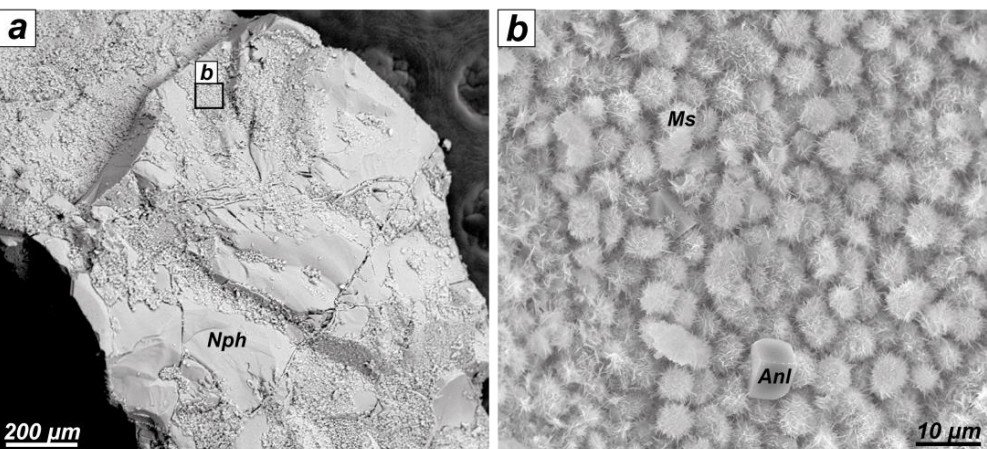

**Figure 5.** *Cont.*

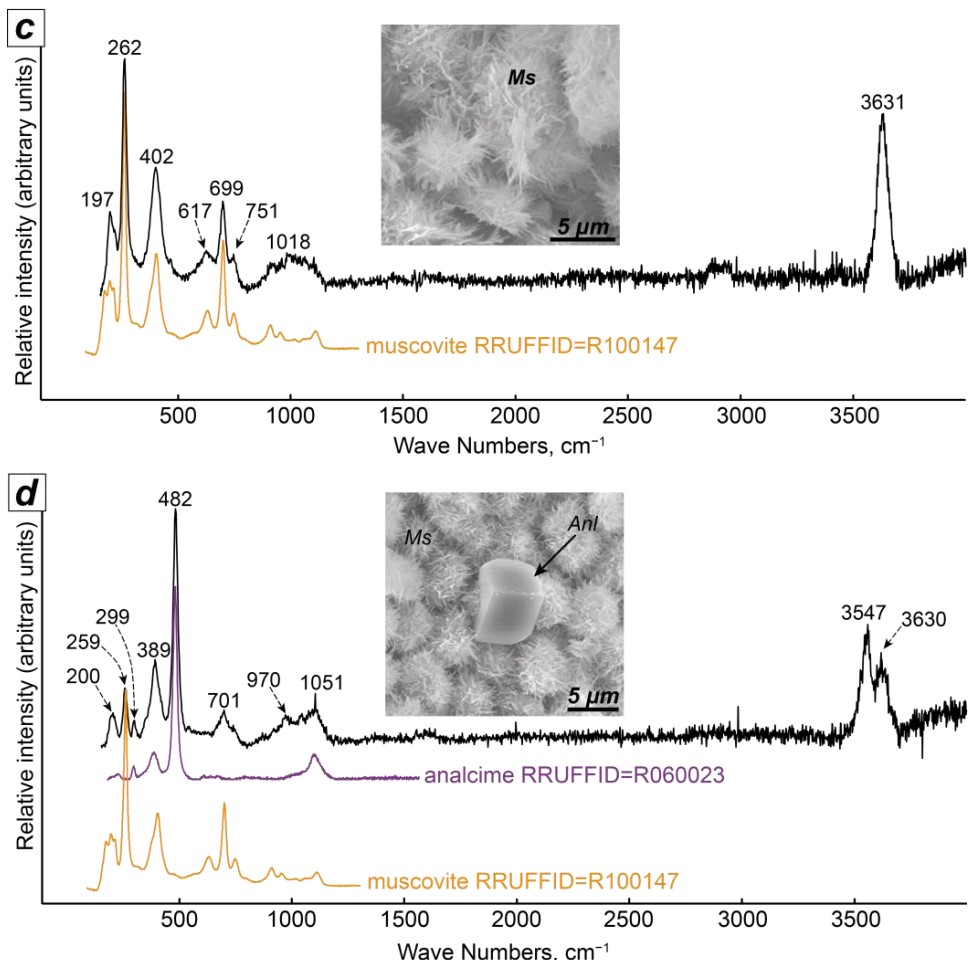

**Figure 5.** Phases precipitated on the surface of nepheline (Nph) grains after experiment with deionized water for 15 days (Aks. 12). (**a**) BSE-image of the nepheline surface covered with newly formed phases; (**b**) detailed BSE-image of the newly formed phases; muscovite (Ms) forms spherulites consisting of small plates, analcime (Anl) forms cubic crystals; (**c**) Raman spectrum of muscovite; (**d**) Raman spectrum of analcime + muscovite. The black lines are the spectra obtained in this study; the colored lines are the spectra from the RRUFF database (https://rruff.info accessed on 19 July 2023). Next to each spectrum is a BSE-image, where an arrow indicates the point of analysis.

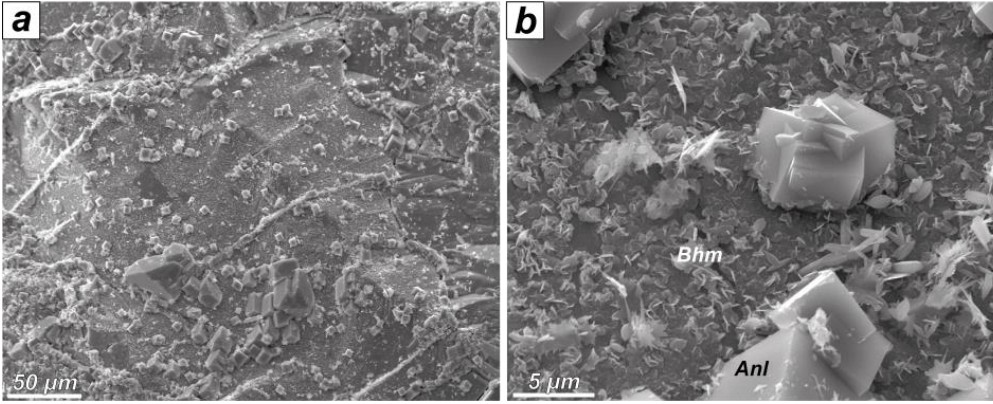

**Figure 6.** *Cont.*

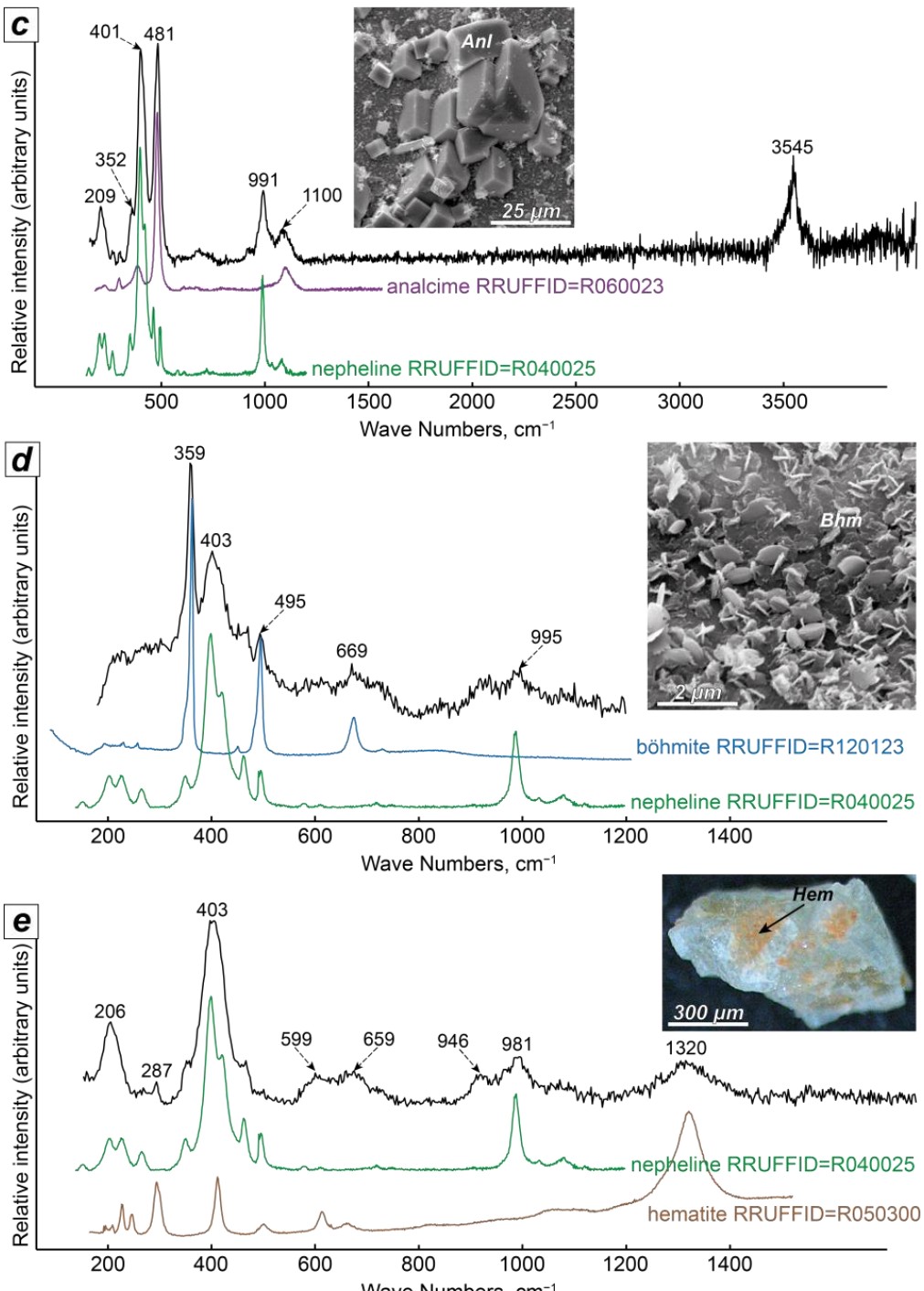

**Figure 6.** Phases precipitated on the surface of nepheline grains after experiment with 0.5 mol/L NaCl solution for 15 days (Aks. 10). (**a**) BSE-image of the nepheline surface covered with newly formed phases; (**b**) detailed BSE-image of the newly formed phases; böhmite (Bhm) forms extremely fine-grained tabular crystals, analcime (Anl) forms cubic crystals; (**c**) Raman spectrum of analcime on the nepheline surface; (**d**) Raman spectrum of böhmite on the nepheline surface; (**e**) Raman spectrum of hematite on the nepheline surface. The black lines are the spectra obtained in this study; the colored lines are the spectra from the RRUFF database (https://rruff.info accessed on 19 July 2023). Next to each spectrum is a BSE-image or photo, where an arrow indicates the point of analysis.

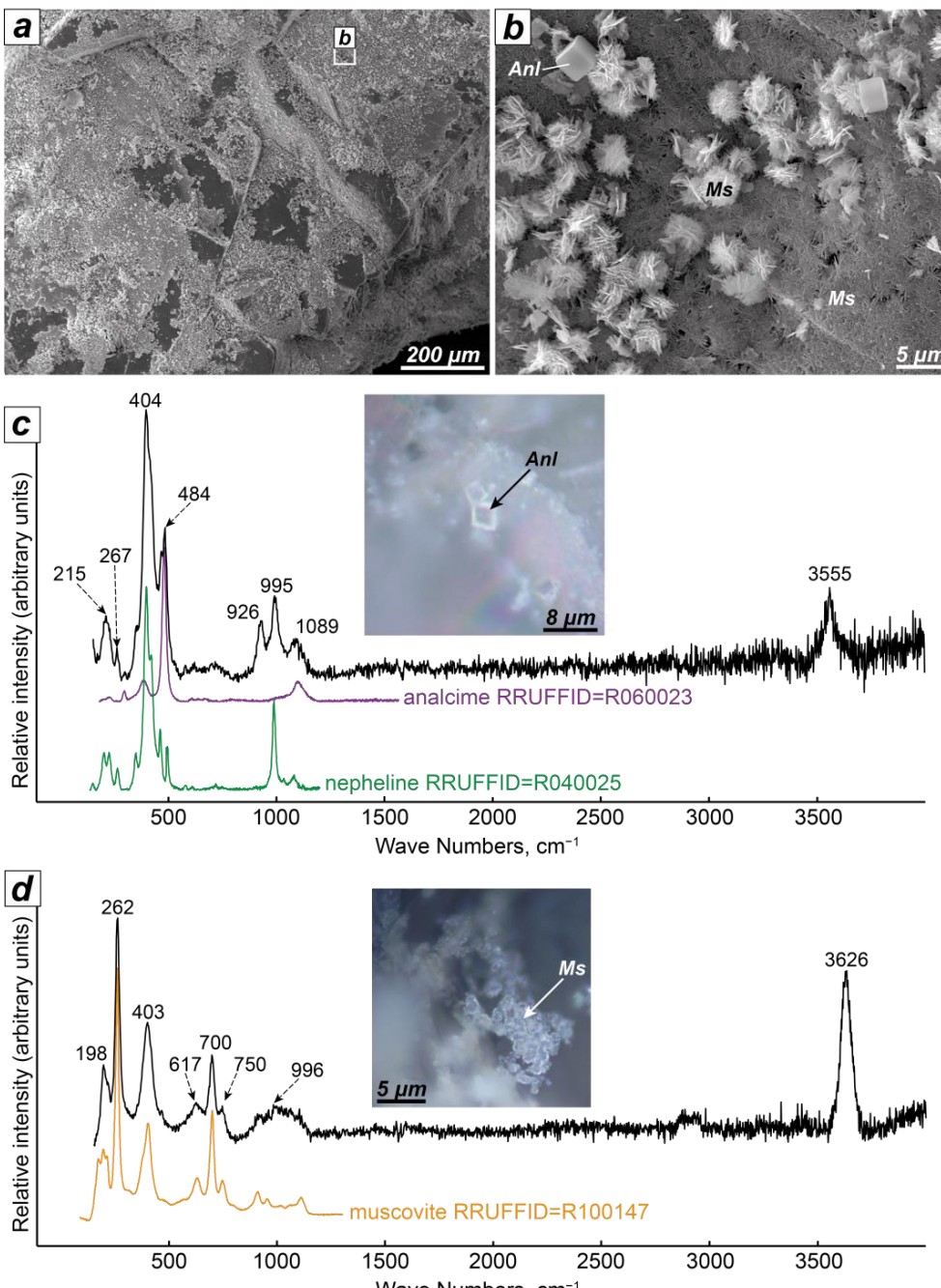

**Figure 7.** Phases precipitated on the surface of nepheline grains after experiment with 0.1 mol/L HCl solution for 15 days (Aks. 9). (**a**) BSE-image of the nepheline surface covered with newly formed phases; (**b**) detailed BSE-image of the newly formed phases; muscovite (Ms) forms spherulites consisting of small plates, analcime (Anl) forms cubic crystals; (**c**) Raman spectrum of analcime on the nepheline surface; (**d**) Raman spectrum of muscovite. The black lines are the spectra obtained in this study; the colored lines are the spectra from the RRUFF database (https://rruff.info accessed on 19 July 2023). Next to each spectrum is a photo in transmitted light, where the point of analysis is indicated by an arrow.

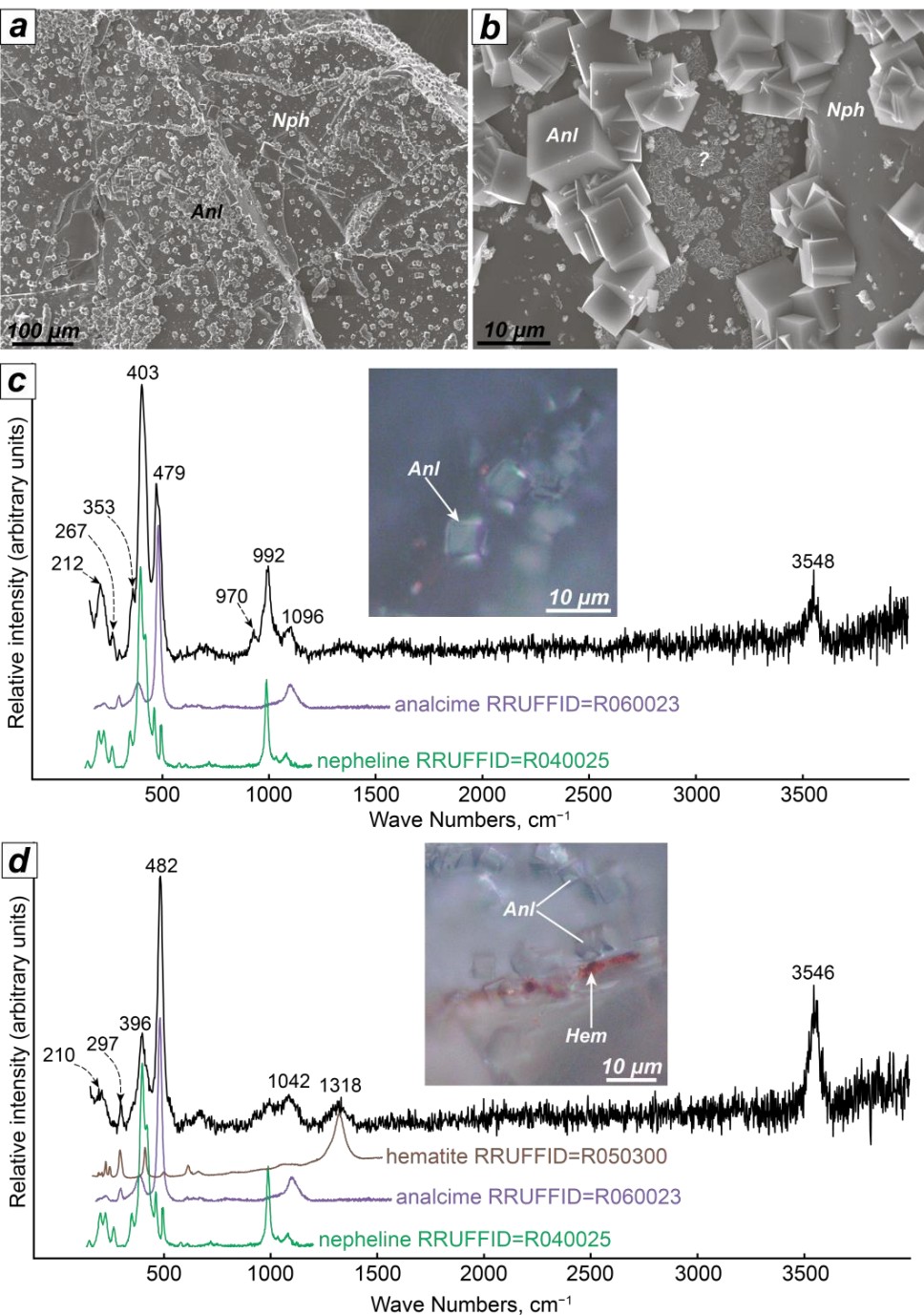

**Figure 8.** Phases precipitated on the surface of nepheline (Nph) grains after experiment with 0.5 mol/L NaOH solution for 15 days (Aks. 11). (**a**) BSE-image of the nepheline surface covered with newly formed phases; (**b**) cubic habit analcime (Anl) crystals on the surface of nepheline (Nph) grains; (**c**) Raman spectrum of analcime on the nepheline surface; (**d**) Raman spectrum of analcime + hematite (Hem) on the nepheline surface. The black lines are the spectra obtained in this study; the colored lines are the spectra from the RRUFF database (https://rruff.info accessed on 19 July 2023). Next to each spectrum is a photo in transmitted light, where an arrow indicates the point of analysis.

**Table 4.** X-ray powder diffraction data (*d* in Å) of precipitated phases, sample Aks. 12 (deionized $H_2O$, 15 days).

| Sample Aks. 12 | | Muscovite ICDD 6-263 | |
|---|---|---|---|
| $I_{meas}$ | $d_{meas}$ | $I$ | $d$ |
| 2 | 9.84 | 95 | 9.95 |
| 6 | 4.94 | 30 | 4.97 |
| 6 | 4.44 | 20 | 4.47 |
| 6 | 3.64 | 18 | 3.73 |
| 6 | 3.32 | 100 | 3.32 |
| 6 | 3.05 | 35 | 2.99 |
| 10 | 2.56 | 55 | 2.57 |
| 8 | 1.496 | 30 | 1.504 |

ICDD—The International Centre for Diffraction Data.

**Table 5.** X-ray powder diffraction data (*d* in Å) of precipitated phases, sample Aks. 10 (0.5 mol/L NaCl, 15 days).

| Sample Aks. 10 | | Analcime ICDD 41-1478 | | Böhmite ICDD 21-1307 | |
|---|---|---|---|---|---|
| $I_{meas}$ | $d_{meas}$ | $I$ | $d$ | $I$ | $d$ |
| 8 | 6.16 | | | 100 | 6.11 |
| 6 | 5.58 | 60 | 5.59 | | |
| 10 | 3.42 | 100 | 3.43 | | |
| 4 | 3.18 | | | 65 | 3.16 |
| 4 | 2.941 | 40 | 2.92 | | |
| 4 | 2.35 | | | 55 | 2.34 |
| 4 | 1.849 | | | 30 | 1.86 |

ICDD—The International Centre for Diffraction Data.

**Table 6.** X-ray powder diffraction data (*d* in Å) of precipitated phases, sample Aks. 9 (0.1 mol/L HCl, 15 days).

| Sample Aks. 9 | | Muscovite ICDD 6-263 | | Analcime ICDD 41-1478 | |
|---|---|---|---|---|---|
| $I_{meas}$ | $d_{meas}$ | $I$ | $d$ | $I$ | $d$ |
| 4 | 9.83 | 95 | 9.95 | | |
| 6 | 5.61 | | | 60 | 5.59 |
| 6 | 4.92 | 30 | 4.97 | | |
| 4 | 3.62 | 18 | 3.73 | | |
| 10 | 3.41 | | | 100 | 3.43 |
| 4 | 3.29 | 100 | 3.32 | | |
| 6 | 3.05 | 35 | 2.99 | | |
| 6 | 2.93 | | | 40 | 2.92 |
| 10 | 2.56 | 55 | 2.57 | | |
| 2 | 1.998 | 45 | 1.993 | | |
| 4 | 1.739 | | | 20 | 1.741 |
| 10 | 1.498 | 30 | 1.504 | | |

ICDD—The International Centre for Diffraction Data.

**Table 7.** X-ray powder diffraction data (*d* in Å) of precipitated phases, sample Aks. 11 (0.5 mol/L NaOH, 15 days).

| Sample Aks. 11 | | Analcime ICDD 41-1478 | |
| --- | --- | --- | --- |
| $I_{meas}$ | $d_{meas}$ | *I* | *d* |
| 8 | 5.63 | 60 | 5.59 |
| 10 | 3.43 | 100 | 3.43 |
| 8 | 2.93 | 40 | 2.92 |
| 2 | 2.50 | 11 | 2.501 |
| 4 | 1.746 | 20 | 1.741 |

ICDD—The International Centre for Diffraction Data.

**Table 8.** Phases precipitated on the surface of partially dissolved nepheline grains under different conditions.

| Sample | Solution | Time, Days | Precipitated Phases |
| --- | --- | --- | --- |
| Aks. 1 | 0.1 mol/L HCl | 1 | no |
| Aks. 2 | 0.5 mol/L NaCl | 1 | no |
| Aks. 3 | 0.5 mol/L NaOH | 1 | no |
| Aks. 4 | deionized $H_2O$ | 1 | no |
| Aks. 5 | 0.1 mol/L HCl | 5 | muscovite |
| Aks. 6 | 0.5 mol/L NaCl | 5 | böhmite |
| Aks. 7 | 0.5 mol/L NaOH | 5 | analcime, hematite |
| Aks. 8 | deionized $H_2O$ | 5 | muscovite |
| Aks. 9 | 0.1 mol/L HCl | 15 | muscovite, analcime, carbon materials |
| Aks. 10 | 0.5 mol/L NaCl | 15 | böhmite, analcime, hematite, carbon materials |
| Aks. 11 | 0.5 mol/L NaOH | 15 | analcime, hematite, carbon materials |
| Aks. 12 | deionized $H_2O$ | 15 | muscovite, analcime, carbon materials |

The presence of carbon materials was established in the samples after 15-day experiments with deionized water, 0.5 mol/L NaCl, 0.1 mol/L HCl, and 0.5 mol/L NaOH by Raman spectroscopy. On the Raman spectra (Figure 9a–c), carbon materials showed characteristic peaks at 1340–1360 cm$^{-1}$ and 1570–1590 cm$^{-1}$ [28,30–32].

According to [28,30–32], graphite single crystal spectrum exhibited a single characteristic line at 1575 cm$^{-1}$, which is designated as G band (after Graphite). Other bands absent from the spectra of graphite single crystals are associated to any type of structural disorder. Thus, a second band at ca. 1360 cm$^{-1}$, occurring together with the G band in spectra of polycrystalline graphites, is designated as D band (after Defects). Thus, in the studied samples, the band at around 1340–1360 cm$^{-1}$ are assigned to D band, and the band at around 1570–1590 cm$^{-1}$ is assigned to G band. We assume that carbon materials cover newly formed minerals such as muscovite and analcime with a thin film.

In some spectra of the studied samples, there is band at 2800–3000 cm$^{-1}$, which corresponds to the C–H-stretching region of organic compounds (Figure 9d).

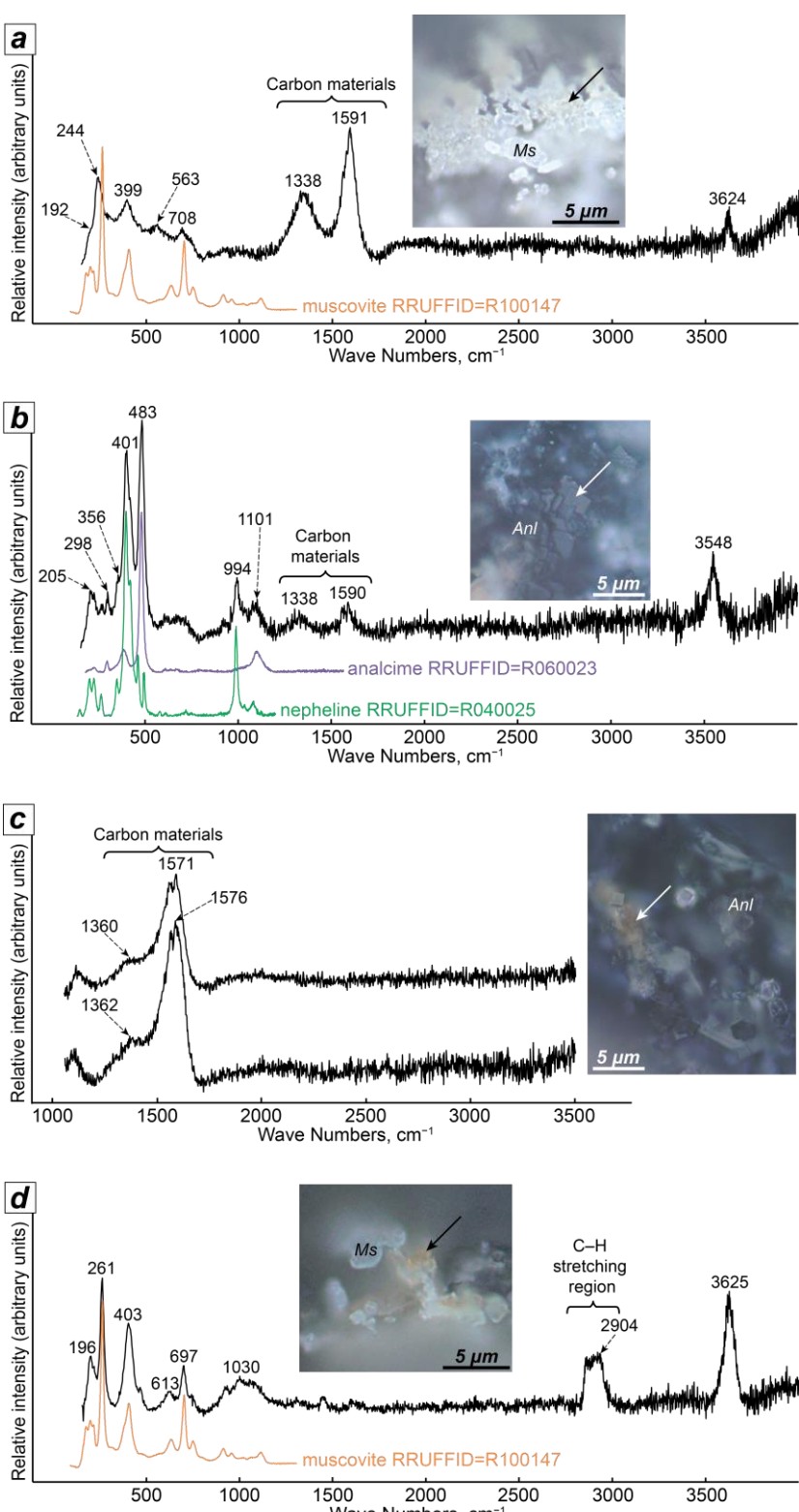

**Figure 9.** Raman spectra of carbon materials and organic compounds. The black lines are the spectra obtained in this study; the colored lines are the spectra from the RRUFF database (https://rruff.info accessed on 19 July 2023). Next to each spectrum is a photo in transmitted light, where an arrow indicates the point of analysis. (**a**) Raman spectrum of muscovite coated with a film (?) of carbon materials; sample Aks. 12 (deionized water, 15 days); (**b**) Raman spectrum of analcime coated with a film (?) of carbon materials; sample Aks. 10 (0.5 mol/L NaCl solution, 15 days); (**c**) Raman spectrum

of analcime coated with a film (?) of carbon materials; sample Aks. 11 (0.5 mol/L NaOH solution, 15 days); (**d**) Raman spectrum of muscovite coated with a film (?) of carbon materials; sample Aks. 12 (deionized water, 15 days).

### 4.4. Solutions after Experiments

Tables A1 and A2 show the chemical composition and pH of both initial solutions and solutions after experiments, and Figure 10 shows the changes in the compositions of the solutions depending on the duration of the experiments.

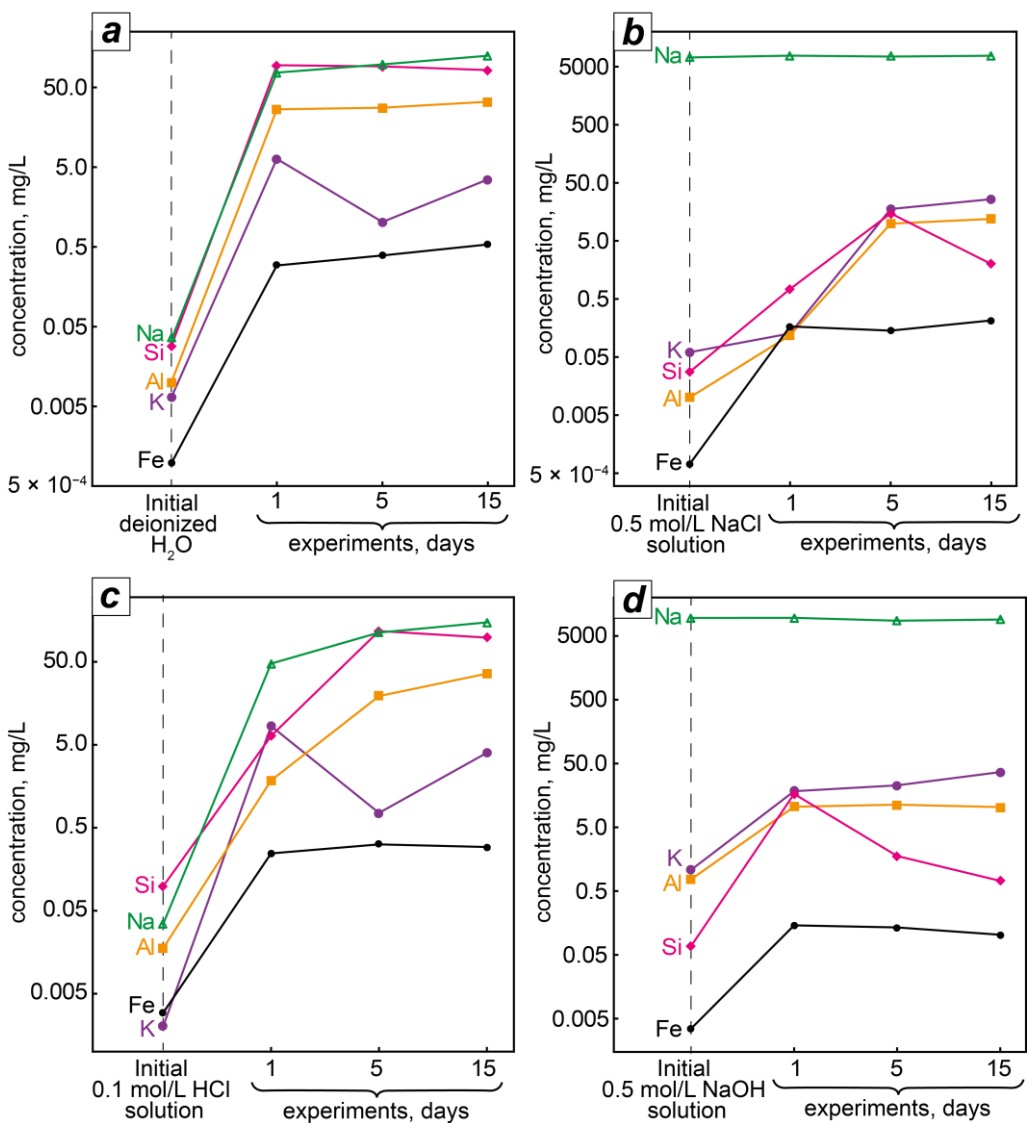

**Figure 10.** Changes in the composition of solutions in experiments on the nepheline dissolution. (**a**) changes in the composition of solutions in experiments with deionized water; (**b**) changes in the composition of solutions in experiments with 0.5 mol/L NaCl; (**c**) changes in the composition of solutions in experiments with 1.5 mol/L HCl; (**d**) changes in the composition of solutions in experiments with 0.5 mol/L NaOH.

When an experiment with deionized water was conducted for 1 day, the concentration of potassium increased from 0.00646 mg/L to 0.006356 g/L (about 1000 times), the Si content increased from 0.0285 mg/L to 0.095 g/L (about 3300 times), and the Al and Na concentrations increased from <0.01 mg/L to 0.02654 g/L (more than 2500 times) and from <0.03 mg/L to 0.07662 g/L (more than 2500 times), respectively. It is impossible to more accurately determine the increase in the concentration of aluminum and sodium, because

the initial concentrations of these elements in deionized water are below the detection limits. However, in general, the ratio of K, Na, Si, and Al in the solution corresponds to the stoichiometry of the dissolving nepheline. With an increase in the duration of the experiment to 5 and 15 days, the potassium and Si concentrations in the solutions decreased, apparently due to the muscovite precipitation. The aluminum concentration remained almost unchanged, while the sodium content increased.

Within 1 day in the experiment with 0.1 mol/L HCl solution, the concentrations of potassium and sodium increased from 0.0020 mg/L to 0.008407 g/L (about 4200 times) and from 0.0344 mg/L to 0.04755 g/L (about 1400 times), respectively, while the concentration of silicon increased only by 65 times (from 0.0982 mg/L to 0.006391 g/L), and the concentration of aluminum increased by 105 times (from 0.0176 mg/L to 0.001859 g/L). With an increase in the duration of the experiment to 5 days, the concentration of silicon and aluminum increased sharply, while the concentration of potassium decreased significantly. In an experiment with 0.1 mol/L HCl solution lasting 15 days, the concentration of potassium in the solution slightly increased, while that of silicon decreased sharply. In experiments with deionized water and 0.1 mol/L HCl, the pH of the solutions increased significantly: from 6.1 (initial deionized water) to 11.2 (after an experiment lasting 15 days) and from 3.0 (initial 0.1 mol/L HCl solution) to 9.7 (after an experiment lasting 15 days).

After the experiment with 0.5 mol/L NaCl for 1 day, the potassium concentration in the solution increased only 2 times, and with an increase in the experiment time to 5 days, it increased about 300 times (from 0.06051 mg/L to 0.01771 g/L) compared to the initial solution. The concentration of aluminum in the experiment lasting 1 day increased by about 12 times, and with an increase in the duration of the experiment to 5 days, the concentration of this component increased by more than 990 times (from <0.01 mg/L to 0.00993 g/L). The concentration of silicon in the experiment lasting 1 day increased by 27 times, while in the experiment lasting 5 days the concentration of silicon turned out to be 542 times higher than in the initial solution (concentration increased from 0.0275 mg/L to 0.01491 g/L). The sodium concentration in the experiment lasting 1 day increased slightly, and in the experiment lasting 5 days it decreased. As a result of experiments with 0.5 mol/L NaCl, the pH of the solutions did not change much.

In an experiment with 0.5 mol/L NaOH lasting 1 day, a sharp increase in the concentration of K, Al, and Si in solution was observed. In 5- and 15-day experiments, the Si concentration consistently decreased, the aluminum concentration remained almost unchanged, and the potassium concentration increased. In the experiment with 0.5 mol/L NaOH lasting 1 day, there was an increase in pH (from 7.8 to 9.8), and in experiments lasting 5 and 15 days, the pH consistently decreased.

In all experiments, the iron content in the solution sharply increases for 1 day, and then (1) gradually increases (in experiments with deionized water), (2) almost does not change (in experiments with 0.1 mol/L HCl) or (3) decreases (in experiments with 0.5 mol/L NaCl and 0.5 mol/L NaOH). The decrease in the iron content in the solutions is apparently associated with the hematite precipitation.

## 5. Discussion

Previous experimental works on nepheline hydrothermal alteration were mainly focused on studying the kinetics, the mechanism of nepheline dissolution, and changes in the composition of solutions as a result of the nepheline dissolution. The results of nepheline decomposition experiments were first published by Morey and Fournier [33]. In this work, the decomposition of potassium-bearing nepheline was investigated by slowly pumping distilled water at 295 °C and 2500 psi over the sample for 135 days. It was found that the decomposition products of nepheline are muscovite, böhmite and analcime, and the ratio of sodium to potassium in the solutions leached from nepheline was greater than the ratio of sodium to potassium in the starting material.

The kinetics of dissolution of nepheline at temperatures of 25, 60 and 80 °C and a pH range of 3 to 11 was studied by Tole and colleagues [34]. According to their data, nepheline

shows first order, congruent dissolution rates, followed by a lowering of the rates due to precipitation of new phases from solutions, initially aluminum hydroxides, and later, as the activity of silica in solution increases, amorphous aluminosilicates. In another experimental work [35], Tole simulated bauxite formation at temperatures of 25, 60, and 80 °C and found that the composition of the products was dependent upon the pH of the solution. At pH 3.0, the reaction products consisted of an aluminum silicate with marked deficiency in Na and K (kaolinite) when compared to fresh nepheline. At pH 5 and 7, the reaction products were predominantly oxides of aluminum, while at pH 9.0 and 11.0, the products consisted of sodium and potassium aluminosilicates.

The mechanism of nepheline dissolution was previously considered in detail by Tole and colleagues [34]. In general, the dissolution process of nepheline is similar to that for other aluminosilicates, such as feldspar [36]. In the process of dissolution, monovalent metal-oxygen bonds break more rapidly than divalent metal-oxygen bonds, which break faster than trivalent metal-oxygen bonds, which break faster than Si-O bonds. The dissolution of an aluminosilicate proceeds by the sequential breaking of metal-oxygen bonds resulting in the release of various metals from the mineral surface. Dissolution is initiated by the removal of the metal having the fastest breaking metal-oxygen bonds. The removal of metals from the structure is coupled to the addition of protons into the mineral structure so the metal removing reactions are metal-proton exchange reactions [37]. Dissolution continues by the successive removal of metals in the order of the relative rates for breaking their corresponding metal-oxygen bonds, until the mineral structure is destroyed.

According to Tole et al. [34], the dissolution of nepheline includes the following sequential steps:

(1)   The $Na^+$ ions at the nepheline surface are rapidly exchanged for $H^+$, forming a surface layer of $HAlSiO_4$. Further absorption of protons results in a positively charged surface $H_2AlSiO_4^+$;

(2)   Si-O-Al bonds break, forming a more open structure, which can allow exchange between protons and the second layer of $Na^+$ ions to take place;

(3)   Disruption of inner Si-O-Al bonds releases the first silicic acid molecule, and $Al^{3+}$ ion.

The details of steps (2) and (3) are pH dependent. For example, at pH = 7.0, the stable aqueous aluminum species is $Al(OH)_4^-$ [38]. Therefore, attack by an oxygen lone pair of electrons (of either $H_2O$ or $OH^-$) on Al and Si atom sites, rather than attack by protons is probably more predominant, and leads to the release of aluminum.

In present work, we have carried out four series of experiments on the dissolution of nepheline under hydrothermal conditions. In experiments with deionized water and 0.1 mol/L HCl solution, muscovite and analcime crystallized, while in experiments with 0.5 mol/L NaCl and 0.5 mol/L NaOH, analcime + böhmite and analcime were precipitated, respectively. In addition, in experiments with 0.5 mol/L NaCl and 0.5 mol/L NaOH, hematite was formed. In all experiments lasting 15 days, the presence of carbon materials was established. With an increase in the duration of the experiment, the number of precipitated phases increased (Table 8).

The composition of the solution after a 1-day experiment with deionized water corresponds to the nepheline stoichiometry (Table A1). Indeed, since nepheline dissolves congruently, the composition of the solution must match the composition of the solute. The Si/Al ratio in nepheline and, respectively, in solution is close to 1. Therefore, minerals with Si/Al close to 1 (kaolinite, $Al_2Si_2O_5(OH)_4$ or muscovite, $KAl_2(AlSi_3O_{10})(OH)_2$) are likely to crystallize from such a solution. Indeed, after experiments with deionized water, tiny plates of muscovite crystallized on the surface of nepheline. The muscovite precipitation led to a decrease in the concentration of K, Si and Al in solution, while sodium accumulated in the solution. Muscovite precipitation can be described by the following schematic reaction:

$$K^+_{(aq)} + 3H_4SiO_{4(aq)} + 3Al(OH)_4^-_{(aq)} + 2H^+_{(aq)} \rightarrow KAl_2(AlSi_3O_{10})(OH)_{2(s)} + 12H_2O \quad (4)$$



As a result of an increase in the sodium content, the solution reached supersaturation with respect to analcime, and small cubic crystals of this mineral crystallized on muscovite (Figure 5b) in accordance with the following scheme:

$$Na^+_{(aq)} + 2H_4SiO_{4(aq)} + Al(OH)_4^-_{(aq)} \rightarrow Na(AlSi_2O_6)\cdot H_2O_{(s)} + 5H_2O \tag{5}$$

The accumulation of sodium in solution during the muscovite crystallization led to an increase in pH. During the subsequent crystallization of analcime, the pH continued to increase. The ratio of Na, Al, and Si in the analcime composition is 1:1:2, and in the dissolving nepheline this ratio is 1.5:2:2. Thus, during the crystallization of analcime, excess sodium and aluminum should accumulate in the solution. Since the solubility of böhmite increases significantly with increasing pH [38,39], excess aluminum did not precipitate as böhmite but remained in solution.

In a 1-day experiment with 0.1 mol/L HCl, a sharp increase in the content of Na and K in relation to Si and Al in solution is observed (Table A2 and Figure 10c). Thus, the stepwise dissolution of nepheline is manifested, when the first stage is the exchange of an alkali metal for a proton [34]. Therefore, sodium and potassium pass into solution earlier than Si and Al. Thus, during the 1-day experiment, a partial dissolution of nepheline in the near-surface layer occurred, namely, the removal of extra-framework cations. Further dissolution of nepheline led to the attainment of supersaturation relative to muscovite first, and then, as sodium content increased, to the supersaturation relative to analcime. As a result, first muscovite and then analcime crystals are precipitated on the surface of nepheline grains (Figure 7a,b).

In experiments with 0.5 mol/L NaCl solution, the dissolution of nepheline was slower than in deionized water or in 0.1 mol/L HCl. The initial sodium concentration in this solution was high, and in order to attain supersaturation with respect to analcime, it was necessary to increase the Al and Si concentrations. After reaching supersaturation, the precipitation of analcime occurred in accordance with the Scheme (4).

Since the initial sodium content in the solution was high, the addition of sodium as a result of the nepheline dissolution did not have a significant effect on pH. The pH value changed from 5.6 to 6.0 (Table A1). In this range of pH, the solubility of böhmite decreases significantly [38,39]; therefore, excess aluminum, which was not included in analcime, crystallized as böhmite. The precipitation of this mineral can be expressed as the following scheme:

$$Al(OH)_4^-_{(aq)} + H^+_{(aq)} = AlO(OH)_{(s)} + 2H_2O \tag{6}$$

Similar to experiments with NaCl, in experiments with 0.5 mol/L NaOH, the dissolution of nepheline was less intense than in in deionized water or in 0.1 mol/L HCl. The initial sodium concentration in this solution was high (Table A2), and in order to achieve supersaturation with respect to analcime, it was necessary to increase the Al and Si concentrations in solution. There was no significant increase in pH with partial dissolution of nepheline and precipitation of analcime, since the initial sodium content was high. Böhmite precipitation, as in experiments with 0.5 mol/L NaCl, apparently did not occur, since the pH values of the solutions corresponded to the region of high böhmite solubility.

In the rocks of the Lovozero massif, intensive secondary alterations of nepheline are observed [12,21,27]. Typical products of such alterations are natrolite and Al-O-H phases (böhmite and/or nordstrandite). In our experiments with 0.5 mol/L NaCl, after partial dissolution of nepheline, analcime and böhmite precipitated. Analcime and natrolite belong to the zeolite group, but differ in the framework type (ANA framework type in analcime; NAT framework type in natrolite according to the International Zeolite Association {http://www.iza-online.org accessed on 19 July 2023}). These minerals are similar in chemical composition, but the Si/Al ratio in the ideal formula of analcime is 2/1 = 2, and in the ideal formula of natrolite it is 3/2 = 1.5. It can be argued that the analcime obtained in experiments is an equivalent of natrolite, which is formed in a natural process. Thus, the

possible precipitation of natrolite upon dissolution of nepheline can be represented as the following scheme:

$$2Na^+_{(aq)} + 3H_4SiO_{4(aq)} + 2Al(OH)_4{}^-_{(aq)} = Na_2(Si_3Al_2)O_{10} \cdot 2H_2O_{(s)} + 8H_2O, \qquad (7)$$

which is similar to the precipitation scheme of analcime (i.e., Scheme (5)).

In addition to böhmite, nordstrandite was found in the products of nepheline alteration in natural samples [12]. Gastuche and Herbillon [40] showed that crystalline $Al(OH)_3$ precipitated from $NaOH$–$AlCl_3$ solutions when no additives were present, but additional chloride slowed down, or prevented, the precipitation of crystalline $Al(OH)_3$ and caused less structurally ordered phases to form. Violante and Huang [41] added NaCl to $NaOH$–$AlCl_3$ solutions and found that higher concentrations of chloride produced smaller amounts of crystalline $Al(OH)_3$ and greater amounts of pseudoböhmite, a gelatinous phase. Only pseudoböhmite was precipitated at very high chloride concentrations. In similar experiments, crystalline $Al(OH)_3$ only slowly replaced the pseudoböhmite after several years of aging [42]. In the pseudoböhmite, the amount of hydroxyl groups and water are both slightly higher than in böhmite. This may explain the slightly larger unit cell, where a small amount of the oxygen atoms has been replaced by hydroxyl groups and maybe even water molecules [43]. Thus, the precipitation of nordstrandite in experiments with 0.5 mol/L NaCl could be inhibited by a high content of chloride in the solution. In the rocks of the Lovozero massif, the content of chlorine in the hydrothermal solution is probably controlled by the intensity of sodalite decomposition, which is commonly associated with nepheline [19,26]. Thus, based on the results of the experiments, it can be concluded that the replacement of nepheline in the rocks of the Lovozero massif with the formation of natrolite and Al-O-H phases occurred under the influence of a high to medium salinity solution at a pH of near 6.

We assume that pressure and temperature do not play a significant role in the alteration of natural nepheline. Similar products of nepheline substitution were obtained in both our experiments and in experiments of Tole and colleagues [34,35]. At the same time, the temperature and pressure differed significantly.

In experiments with 0.5 mol/L NaCl and 0.5 mol/L NaOH, precipitation of hematite was observed (Figures 6e and 8d), accompanied by a decrease in the concentration of iron in solutions (Tables A1 and A2). Considering that, in addition to ferric iron, nepheline also contains ferrous iron (Table 3), it should be noted that when nepheline was dissolved, iron was oxidized in accordance with the schemes:

$$Fe^{2+} + OH^- \rightarrow Fe^{3+} + O^{2-} + 0.5H_2 \qquad (8)$$

$$\text{or } Fe^{2+} + H_2O \rightarrow Fe^{3+} + O^{2-} + H_2. \qquad (9)$$

Both in natural pseudomorphized nepheline and among the precipitated phases in our experiments, carbon materials are present (Figures 1c–e and 9). At the same time, we found only small methane inclusions in the unaltered nepheline. Carbon matter in nepheline from the rocks of the Khibiny massif was previously studied by S. Ikorsky [23]. He extracted carbon matter from nepheline and other minerals in Soxhlet apparatus by chloroform and assumed that (1) carbon matter was contained in the nepheline-hosted fluid inclusions or (2) carbon matter was dispersed within nepheline grains. We assume that the carbon materials were not formed in the course of experiments (or in the natural process of nepheline alteration), but was released from nepheline when it was (partially) dissolved. We also assume that carbon materials were contained in the nepheline-hosted inclusions along with methane, but due to the small size of such inclusions, it was not detected by Raman spectroscopy.

## 6. Conclusions

Observations in natural samples and experimental studies have shown that the nature of the products of nepheline alteration is determined by the sodium and chlorine concentrations in the hydrothermal solution, as well as pH.

(1)   *Role of sodium.* At low sodium content in the hydrothermal solution, muscovite is the main product of nepheline alteration. Increasing the sodium content of the hydrothermal solution leads to the precipitation of zeolites (analcime or natrolite) and Al-O-H phases.

(2)   *Role of chlorine.* Depending on the chlorine content, aluminum precipitates either as nordstrandite, or as böhmite, or as nordstrandite + böhmite association.

(3)   *Effect of pH.* At high pH (>9.0), the solubility of böhmite increases, this mineral does not precipitate, and aluminum remains in solution.

Thus, nepheline alteration in the rocks of the Lovozero massif with the formation of natrolite and Al-O-H phases occurred under the influence of a high to medium salinity solution at a pH of near 6.

**Author Contributions:** Conceptualization, J.A.M. and G.O.K.; methodology, G.O.K.; software, A.A.K.; validation, E.A.S., Y.A.P. and A.A.K.; investigation, E.A.S., Y.A.P. and A.A.K.; resources, J.A.M.; data curation, J.A.M.; writing—original draft preparation, J.A.M.; writing—review and editing, G.O.K., E.A.S. and Y.A.P.; visualization, A.A.K. and J.A.M. All authors have read and agreed to the published version of the manuscript.

**Funding:** This research was funded by Russian Science Foundation, project No. 21-47-09010.

**Data Availability Statement:** Not applicable.

**Acknowledgments:** We are grateful to the reviewers who helped us improve the presentation of our results.

**Conflicts of Interest:** The authors declare no conflict of interest.

## Appendix A

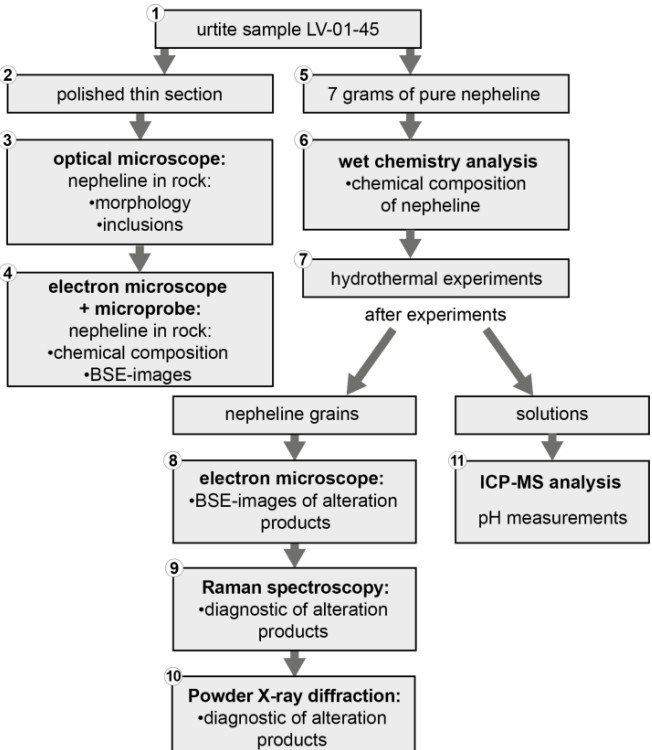

**Figure A1.** General scheme of the study.

**Table A1.** Chemical composition (mg/L) and pH of deionized $H_2O$, 0.5 mol/L NaCl solution and solutions after experiments.

| Component, mg/L | Initial Deionized $H_2O$ | After Experiments | | | Initial 0.5 mol/L NaCl Solution | After Experiments | | |
|---|---|---|---|---|---|---|---|---|
| | | Aks. 4 (1 Day) | Aks. 8 (5 Days) | Aks. 12 (15 Days) | | Aks. 2 (1 Day) | Aks. 6 (5 Days) | Aks. 10 (15 Days) |
| K | 0.00646 | 6.356 | 1.027 | 3.510 | 0.06051 | 0.127 | 17.71 | 26.12 |
| Al | <0.01 | 26.54 | 27.86 | 33.03 | <0.01 | 0.119 | 9.930 | 11.93 |
| Si | 0.0285 | 95.00 | 91.74 | 82.48 | 0.0275 | 0.741 | 14.91 | 1.999 |
| Na | <0.03 | 76.62 | 97.12 | 125.0 | 7197 | 7728 | 7471 | 7719 |
| Fe | 0.00097 | 0.293 | 0.390 | 0.537 | 0.00071 | 0.168 | 0.143 | 0.214 |
| pH | 6.1 | 7.1 | 9.8 | 11.2 | 5.6 | 5.8 | 5.6 | 6.0 |

**Table A2.** Chemical composition (mg/L) and pH of 0.1 mol/L HCl solution, 0.5 mol/L NaOH solution and solutions after experiments.

| Component, mg/L | Initial 0.1 mol/L HCl Solution | After Experiments | | | Initial 0.5 mol/L NaOH Solution | After Experiments | | |
|---|---|---|---|---|---|---|---|---|
| | | Aks. 1 (1 Day) | Aks. 5 (5 Days) | Aks. 9 (15 Days) | | Aks. 3 (1 Day) | Aks. 7 (5 Days) | Aks. 11 (15 Days) |
| K | 0.0020 | 8.407 | 0.754 | 3.997 | 1.073 | 18.42 | 22.72 | 36.69 |
| Al | 0.0176 | 1.859 | 19.33 | 35.99 | 0.764 | 10.51 | 11.31 | 10.37 |
| Si | 0.0982 | 6.391 | 117.3 | 97.92 | 0.069 | 16.67 | 1.776 | 0.723 |
| Na | 0.0344 | 47.55 | 112.5 | 149.4 | 9586 | 9627 | 8673 | 9044 |
| Fe | 0.00293 | 0.245 | 0.315 | 0.393 | 0.003 | 0.145 | 0.135 | 0.103 |
| pH | 3.0 | 3.6 | 6.2 | 9.7 | 7.8 | 9.8 | 9.3 | 9.1 |

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
