# Peer review of "Experimental Modeling of Natural Processes of Nepheline Alteration"

_minerals, doi:10.3390/min13091138_

Round 1

Reviewer 1 Report

An interesting experimental work was carried out. The authors studied the products of nepheline change under the influence of 0.5 mol/L NaCl, 0.1 mol/L HCl, 0.5 mol/L NaOH and deionized water solutions. Extensive qualitative work has been done to study the products of nepheline change under experimental conditions. A wide range of methods were used to investigate the results of experiments. The chemical compositions of nepheline was measured by wet chemical analysis and electrone microprobe analysis, gas inclusions in nepheline was studied by Raman spectroscopy. Also the products of nepheline alteration was studied by electron microscope, X-ray powder diffraction and by Raman spectroscopy. The composition of the solutions after experiments was studied using ICP MS analysis. It was shown, that the ratio of K, Na, Si, and Al in the solution generally corresponds to the stoichiometry of the dissolving nepheline, but with an increase in the duration of the experiment to 5 and 15 days, the potassium and Si concentrations in the solutions decreased, apparently due to the muscovite precipitation.

The authors conducted a detailed analysis of the behavior of elements in the solution during the dissolution of nepheline, and also studied the mechanism of change in the composition of solutions compared to natural processes on an example of Lovozero massif. This interesting experimental work certainly deserves publication in Minerals.

Author Response

Thank you very much for your appreciation of our work. We hope that the proposed manuscript will be of interest to readers of Minerals.

Reviewer 2 Report

This  manuscript is interesting and well written and presents a novel investigation on hydrothermal alteration of natural nepheline. The experimental results presented here are well supported with the data and methods used and offer an explanation to understand the condition in which such hydrothermal processes may occur. I am convinced that the manuscript will be of interest for Minerals readers. However, it requieres moderate/minor revision before being accepted for publication. A Figure explaining the structure of the Lovozero massif and its location should be included at the beginning. The role of pressure, which is not considered, in the experiments should be discussed even if it is assumed to be not relevant in such hydrothermal processes. An annotated pdf file is attached with some more comments

English text is correct but some  expressions could be improved. I recommend to check the English text once more.

Author Response

This  manuscript is interesting and well written and presents a novel investigation on hydrothermal alteration of natural nepheline. The experimental results presented here are well supported with the data and methods used and offer an explanation to understand the condition in which such hydrothermal processes may occur. I am convinced that the manuscript will be of interest for Minerals readers. However, it requieres moderate/minor revision before being accepted for publication.

We are very grateful for the detailed comments, which greatly improved the manuscript.

Point 1. A Figure explaining the structure of the Lovozero massif and its location should be included at the beginning.

Response 1. Figure 1 was added - the geological scheme of the Lovozero massif.

Point 2. The role of pressure, which is not considered, in the experiments should be discussed even if it is assumed to be not relevant in such hydrothermal processes.

Response 2. We've added the following text to the "Experimental Conditions" section: "During the experiments, the pressures in the autoclaves were autogenous pressures" (lines 154-155). A discussion of the role of pressure has also been added to the "Discussion" section (lines 515-518).

Point 3. An annotated pdf file is attached with some more comments.

Response 3. In the attached pdf file, we have answered all the comments of the reviewer.

We checked the English language once more and made some corrections.

Reviewer 3 Report

The paper presents a captivating and innovative approach, even though it draws from existing literature in certain sections. The abundance of data is noteworthy, and the experimental segment is both extensive and engrossing. To enhance accessibility, consider consolidating the information using synoptic tables. Additionally, relocating a portion of the methodology to the appendix could streamline comprehension for readers. While we've only suggested one additional pertinent reference concerning this rock type, the work essentially meets publication standards with minor revisions. However, there's potential for improvement by condensing certain sections and providing more intricate conclusions. For detailed feedback, please refer to the attached text.

Author Response

The paper presents a captivating and innovative approach, even though it draws from existing literature in certain sections. The abundance of data is noteworthy, and the experimental segment is both extensive and engrossing. To enhance accessibility, consider consolidating the information using synoptic tables. Additionally, relocating a portion of the methodology to the appendix could streamline comprehension for readers. While we've only suggested one additional pertinent reference concerning this rock type, the work essentially meets publication standards with minor revisions. However, there's potential for improvement by condensing certain sections and providing more intricate conclusions. For detailed feedback, please refer to the attached text.

Response. Many thanks for the detailed comments, which helped us improve the manuscript significantly. We have changed the structure of the manuscript in accordance with the recommendations of the reviewer. In the attached file, we have answered all the comments of the reviewer.
